# Human Caspase 12 Enhances NF-κB Activity through Activation of IKK in Nasopharyngeal Carcinoma Cells

**DOI:** 10.3390/ijms22094610

**Published:** 2021-04-28

**Authors:** Shu-Er Chow, Huei-Tzu Chien, Wing-Keung Chu, Victor Lin, Tzu-Hsiu Shen, Shiang-Fu Huang

**Affiliations:** 1Department of Otolaryngology, Head and Neck Surgery, Chang Gung Memorial Hospital, Taoyuan 333, Taiwan; chowse@mail.cgu.edu.tw; 2Department of Nature Science, Center for General Studies, Chang Gung University, Taoyuan 333, Taiwan; smartchuy@gmail.com (W.-K.C.); pallashen@gmail.com (T.-H.S.); 3Department of Nutrition and Health Sciences, Chang Gung University of Science and Technology, Taoyuan 333, Taiwan; kathy.htchien@gmail.com; 4Department of Medicine, Chang Gung University, Taoyuan 333, Taiwan; victorlyx96@gmail.com; 5Graduate Institute of Clinical Medical Sciences, Chang Gung University, Taoyuan 333, Taiwan

**Keywords:** Caspase-12, inflammation, IKK, IκBα, NF-κB, human nasopharyngeal carcinoma

## Abstract

Human nasopharyngeal carcinoma (NPC) is a highly invasive cancer associated with proinflammation. Caspase-12 (Casp12), an inflammatory caspase, is implicated in the regulation of NF-κB-mediated cellular invasion via the modulation of the IκBα protein in NPC cells. However, the effect mechanisms of Casp12 need to be elucidated. NPC cells were transfected with the full length of human Casp12 cDNA (pC12) and the effect of human Casp12 (hCasp12) on the NF-κB activity was investigated. We found ectopic expression of hCasp12 increased the NF-κB activity accompanied by an increased p-IκBα expression and a decreased IκBα expression. Treatment of BMS, a specific IKK inhibitor, and pC12-transfected cells markedly decreased the NF-κB activity and ameliorated the expression level of IκBα reduced by hCasp12. Co-immunoprecipitation assays validated the physical interaction of hCasp12 with IKKα/β, but not with NEMO. Furthermore, the NF-κB activity of ΔCasp12-Q (a mutated catalytic of hCasp12) transfected cells was concentration-dependently induced, but lower than that of hCasp12-transfected cells. Importantly, the hCasp12-mediated NF-kB activity was enhanced by TNFα stimulation. That indicated a role of the catalytic motif of hCasp12 in the regulation of the NF-κB activity. This study indicated hCasp12 activated the NF-κB pathway through the activation of IKK in human NPC cells.

## 1. Introduction

Human nasopharyngeal carcinoma (NPC) is a dominant head and neck cancer in southern China and South East Asia. One of the most striking and consistent characteristics of NPC is the presence of a very abundant leucocyte infiltrate mainly containing T-lymphocytes, macrophages, suggesting an important link of the pro-inflammatory factors for the processes of carcinogenesis. Chronic inflammation is a common and high impacting feature in metabolic diseases and cancer metastasis [1,2]. The inflammatory caspase is a category of the caspase family that functions in both apoptosis and inflammatory cytokine production, engaging in the switches of inflammation [3]. Abnormal activation of nuclear factor-κB (NF-κB) signaling has been an indicator of chronic inflammatory diseases, including cancer. However, the inflammatory role involved in the pathophysiological processes of NPC remains to be investigated.

Caspase 12 (Casp12), an inflammatory caspase, is implicated in the modulation of inflammatory signaling. In humans, a single nucleotide polymorphism of the human Casp12 (hCasp12) gene generates a truncated or a full-length protein [4]. The distribution of hCasp12 is cell-type specific and a selective advantage in humans [5]. Casp12 plays an important role in apoptosis and inflammation [6]. An induction of hCasp12 expression is presented in human proximal tubule of kidney and human cancers [7,8,9]. The hCasp12-mediated apoptotic signaling pathway is detected in NPC and hepatoma cells [8]. Induction of Casp12 by PMA (a PKC analogue) leads to cell invasion in NPC cells [10]. Ectopic murine Casp12 protein is shown to decrease the IκBα protein and significantly increase the NF-κB activity in NPC cells [10]. Moreover, downregulation of Casp12 impairs the cancer cell invasion via inactivation of NF-κB [10]. A recent study indicated that knockout of Casp12 failed to induce Casp1 activation, IL-1β, and IL-18 production [11]. These observations suggested a role of Casp12 in NF-κB -mediated cell invasion. The inflammatory role of Casp12 in NPC cells needs to be investigated.

An inflammatory microenvironment and oncogenic mutations in human cancer pathology often induce chronic inflammation, leading to an exhibition of constitutive NF-κB activity [2]. NF-κB (p65/p50) is restricted to the cytoplasm by the inhibitor of the κB (IκB) family [12]. The IκB kinase (IKK) complex is the signal integration hub for NF-κB activation [13]. The IKK complex contains two catalytic subunits, the IKKα/β and a regulatory subunit, NF-κB essential modulator (NEMO) [14]. The activation of NF-κB mostly occurs via the IKK-mediated phosphorylation of IκBα molecules [14]. The IκBα phosphorylation induces its degradation that leads to the p65 nuclear translocation. In addition, IκBα activity is also regulated by caspases cleavage [15,16]. Cleavage of IκBα by caspase creates a N-terminal truncated form, thereby resulting in the activation of NF-κB [16]. Induction of Casp12 is implicated in cancer cell invasion after the proinflammation stimuli [10]. This study indicated the alternative effect of hCasp12 on the activation of NF-κB through the physical interaction of IKKα/β protein in NPC cells.

## 2. Results

### 2.1. Ectopic hCasp12 Induced the Decreased IκBα Protein and Increased Expression of pIκBα

To investigate the effect of hCasp 12 on the IκBα expression, pC12 was transfected to NPC cells for 24 h and the cell lysates were subjected to western blot for detection of IκBα, IKKα/β, and NEMO proteins. As shown in Figure 1A, the endogenous Casp12 (~50 kDa) and the transfected hCasp12 (pC12 contained GFP, ~65 kDa) were detected. Ectopic expression of hCasp12 concentration-dependently decreased the level of IκB protein, but did not change the levels of NEMO or IKKα/β proteins. hCasp12 also increased the phosphorylation level of IκBαprotein (p-IκBα), but did not change the phosphorylation level of p65 protein (p-p65) (Figure 1B).

The effect of hCasp12 on p65 nuclear translocation was examined via p65 western blot detection in hpC12 transfected NPC cell lysates. As shown in Figure 1C, ectopic expression of hpC12 markedly increased the p65 protein level in nuclear fraction lysates, but it did not affect the p65 protein level in total cell lysates. The results indicated that ectopic hCasp12 increased the p-IκBα level, but decreased the IκBα protein, indicating induction of p65 nuclear translocation.

### 2.2. The Casp12-Mediated NF-κB Activity Regulated by the IKK Complex

The activation of NF-κB mostly occurs via the IKK-mediated phosphorylation of IκBα molecules [17]. We examined the role of IKK on the activation of hCasp12-mediated NF-κB, NPC cells transfected with pC12, and p65 reporter plasmid in the presence of BMS, a specific IKK inhibitor, for 24 h. As shown in Figure 2A, transfection of pC12 markedly enhanced the NF-κB activity. Treatment of BMS concentration-dependently decreased the hCasp12-mediated NF-κB activity. The effect of IKK on the level of the IκBα protein decreased by hCasp12 was examined. As shown in Figure 2B, co-incubation of BMS markedly ameliorated the decreased IκBα protein induced by hCasp12. The results indicated hCasp12-mediated NF-κB activity was involved in the activation of IKKs.

### 2.3. The Interaction of hCasp12 with IKK Complex

To examine whether or not hCasp12 and IKK complex interacted physically, hCasp12 was immunoprecipitated from the total cell lysates of the hpC12-transfected cells, then the levels of IKKα/β were validated by western blot (Figure 3A). Along with the increased concentration of hpC12 transfection in cells, the immunoprecipitates contained high levels of IKKα and IKKβ in the pC12-transfected cells (Figure 3A). The protein interaction between Casp12 and IKK was further examined by the co-immunoprecipitation assay with anti-IKKα antibody (Figure 3B). The immunoprecipitates containing higher levels of hCasp12, but lower levels of NEMO were detected in the pC12-transfected cells than the control cells. Furthermore, the interaction between hCasp12 and NEMO was examined by co-immunoprecipitation with anti-NEMO antibody. As shown in Figure 4, the immunoprecipitates containing lower levels of IKKα/β were detected in the pC12-transfected cells than the control cells. The results indicated the physical interaction of hCasp12 and IKKα/β.

### 2.4. The Activity of Casp12 Is Involved in the NF-κB Activity

The activation of NF-κB can be mediated, at least partially, by interaction motifs present in the prodomains of specific caspases [18]. Z-Ala-Thr-Ala-Asp(OMe)-FMK (Z-ATAD-fmk), a synthetic peptide inhibitor irreversibly binds to the active cleft of hCasp12. To delineate the activity of hCasp12 on the NF-κB activity, NPC cells were treated with Z-ATAD-fmk and cotransfected with pC12 and NF-κB reporter plasmid. As shown in Figure 5, the NF-κB activity was enhanced after transfection with pC12 for 24 h. Co-incubation with Z-ATAD-fmk markedly suppressed the NF-κB activity induced by pC12 concentration- dependently. The active site of hCasp12 in the modulation of NF-κB was further examined. We constructed ΔCasp12-Q (VC12) in which the QACRG pentapeptide motif containing the active site cysteine was replaced [19]. The NF-κB activity was determined after 24 h transfection of NPC cells with pC12/VC12 and NF-κB reporter plasmid. As shown in Figure 6A, the NF-κB activity was markedly decreased in the VC12-transfected cells compared to the pC12-transfected cells. The role of VC12 on the regulation of NF-κB was further examined in the presence of Z-ATAD-fmk for 24 h. As shown in Figure 6B, Z-ATAD-fmk markedly decreased the NF-κB activity of pC12-transfected cells and did not change the NF-κB activity of VC12-transfected cells. The data suggested the active site of hCasp12 is important for the regulation of NF-κB activity.

### 2.5. The Casp12-Mediated NF-κB Activity Was Enhanced by TNFα Stimuli

Tumor Necrosis Factor (TNF) is one of the most potent physiological inducers of the nuclear transcription factor NF-κB [20]. To evaluate a critical role of the Casp12 in response to the inflammatory stimuli, pC12/VC12-transfected cells were treated with TNF-α (10 ng/mL) for 6 h and the cell lysates were used to detect the NF-κB activity. As shown in Figure 7, Treatment with TNF-α induced the NF-κB activity in the control-group cells and markedly enhanced the NF-κB activity in the pC12- or VC12-transfected cells. However, the NF-κB activity in the VC12-transfected cells was markedly lower than that in pC12-transfected cells after stimulation of TNF-α. The data indicated an inflammation role of hCasp12 in NPC cells.

## 3. Discussion

Chronic inflammation is a common and high impacting feature in metabolic diseases and cancer metastasis [1,2]. This study investigated the inflammatory role of hCasp12 in the modulation of NF-kB activity. We found ectopic hCasp12 increased the NF-κB activity accompanied by the activation of IKK complex and an increased level of p-IκBα protein. We found the physical interaction of hCasp12 with IKKα/β, but not with NEMO, led to the activation of IKKs. The active site of hCasp12 played an important role in the modulation of NF-κB activity. Importantly, the NF-κB activity induced by hCasp12 was enhanced after stimulation by TNFα. This study indicated the molecular mechanism of hCasp12 in the proinflammatory response and might have a targeting cancer therapeutic potential.

Casp12 has unique characteristics that are not mutual to other family members of inflammatory caspases. The functional form appears to be confined to people of African descent and is linked with susceptibility to sepsis [4]. Contrary to other inflammatory caspases, Casp12 is proposed to be a negative regulator of inflammatory responses [4]. It indicates people carrying the functional gene have decreased responses to bacterial molecules such as lipopolysaccharide [21], suggesting the anti-inflammation characteristics of Casp12. On the contrary, Casp12-deficiency mice suppressed inflammatory responses by abrogating Casp11 expression and by inhibiting Casp1 activation [11], suggesting the role of Casp12 in the activation of inflammation. Another study indicated a critical role of murine Casp12 on the activation of NF-κB is through the degradation of IκBα in NPC cells [10]. The study suggests Casp12 has a link between inflammatory and aggressive invasion. Recent studies indicated induction of Casp12 in the pathological condition included kidney cells and cancer cells [7,10,22]. We used the human cDNA to explore the physiological significance of hCasp12 in human NPC. In line with the previous finding, this study indicated the IKKs activated by hCasp12 on the NF-kB activation in NPC cells and the results suggested its regulator role in inflammation.

Ectopic hCasp12 induced the levels of p-IκBα and decreased IκBα protein that led to the activation of NF-κB pathway (Figure 1 and Figure 2). IKK complex is a known main regulator of IκBα. We investigated the potential interaction between hCasp12 and IKK complex. The incubation of pC12-transfected NPC cells with IKK inhibitor (BMS) rescued the IκBα protein level reduced by hCasp12 (Figure 2). The result revealed that IKK might be a signal hub molecule for hCasp12-mediated NF-κB activation. We further verified the protein interaction between hCasp12 and IKK complex by co-immunoprecipitation of pC12-transfected NPC cells with anti-Casp12 and anti-IKKβ antibodies (Figure 3). However, the physical interaction between NEMO and hCasp12 was not detected in the immunoprecipitation of anti-NEMO antibody (Figure 4). A recent study that demonstrates Casp12′s inhibitory effect on NEMO in association with the IKK complex and inhibits the activation of NF-kB in HEK293T cells [23]. In contrast to the study, we indicated Casp12 functioned as an activator of NF-kB by the activation of the IKKα/β complex in NPC cells. The role of hCasp12 in the inflammatory response might be in a cellular context-dependent manner. In response to a wide variety of cellular stimuli, NF-kB becomes active through two signaling pathways [24]. The canonical pathway of NF-kB activation depends on the IKK, which contains two catalytic subunits, IKKα and IKKβ, and a regulatory subunit, NEMO. Distinctly, the noncanonical NF-kB pathway is regulated by an IKKα homodimer. The data in Figure 2, Figure 3 and Figure 4 demonstrate that BMS inhibited the Casp12 mediated-NF-kB signaling and ameliorated the IkBα protein reduced by hCasp12 (Figure 2). The activation of IKKs induced by hCasp12 was through physically interacted with IKKα/β, but not with NEMO (Figure 3 and Figure 4). The results suggested the inflammatory effect of Casp12 functioned through an alternative pathway of NF-kB signaling.

It has been shown that the role of murine Casp12 activity on the activation of NF-κB was involved in the upregulation of MMP-9 [10]. This study also indicated the hCasp12 activity marked a role in modulating NF-κB activation, which was abrogated by treatment with hCasp12 inhibitor Z-ATAD-fmk (Figure 5). We examined the dependency of hCasp12′s catalytic site in limited NF-κB activation by VC12-transfected NPC cells, which expressed mutant hCasp12 that mutated the catalytic site (Figure 6). The NF-κB activity was concentration-dependently enhanced in VC12-transfected cells that had a lower level than that in pC12-transfected cells. Interestingly, the NF-κB activity of the VC12-transfected cells was similar to that of pC12 and Z-ATAD-fmk treated cells (Figure 6B), indicating the limited activation of NF-κB by mutant hCasp12. The active site cleft of hCasp12 bound to the tetrapeptide inhibitor Z-ATAD-fmk irreversibly might decrease the activity of hCasp12 or disrupt the physical interaction with IKK. In the respect of hCasp12-mediated IKK activation, at least partially, the physical interaction of hCasp12 and IKKs might be associated with the catalytic motifs of hCasp12. However, the role of catalytic motif of hCasp12 needs to be investigated.

NF-κB target inflammation not only increases the production of inflammatory cytokines, chemokines, and adhesion molecules, but also regulates the cell proliferation, apoptosis, morphogenesis, and differentiation [25]. This study indicated the NF-κB activity was increased after transfection of ectopic expression of pC12/VC12. TNF-α, one of the inflammatory cytokines, enhanced the pC12/VC12-mediated NF-κB activity (Figure 7). This suggested a critical role of Casp12 in inflammation. The activation of IKK complex is responsible for the TNF-α induced phosphorylation of IkBα [20]. The p-IkBα is a signal for its degradation. The enhanced effect induced the hCasp12 and the TNF-α stimulation markedly increased the NF-κB activity (Figure 7). Elevated NF-κB activity was associated with tumor resistance to anticancer therapy, as well as to TNF-α-induced apoptosis, which might help these cells evade immune surveillance. Thus, the interaction of TNF-*α* and hCasp12/VC12 may contribute to the activation of the NF-κB pathway associated with tumor cell invasion and metastasis.

Emerging evidence indicates the important role of caspases as mediators or regulators of NF-κB signaling, which play a key role in inflammation and immunity. As inflammation underlies a wide variety of physiological and pathological processes, we attempted to identify the physiological function effects of hCasp12 on the activation of NF-κB associated with cancer progression. This study indicated hCasp12/IKK might act as a master switch in establishing an intricate link between inflammation and cancer. Moreover, this study indicated the functional effects of hCasp12 on the activation of NF-κB. Furthermore, this study indicated the interactions between pC12/VC12 and TNF-α had a key mediator of inflammation in NPC cells. Understanding the mechanisms underlying hCasp12-mediated inflammation might reveal new therapeutic targets for cancer prevention and treatment.

## 4. Materials and Methods

### 4.1. Cell Culture and Reagents

NPC076 and NPC039 are the isolated nasopharyngeal squamous carcinoma cell lines used in this study [10]. The two cell lines were cultured in DMEM/F-12 (Invitrogen, Carlsbad, CA, USA) supplemented with 5% FBS at 37 °C under 5% CO_2_/95% air. The cell lines were incubated and conditioned with the following reagents for experiment control: *Z*-ATAD-fmk (1–3 μM, BioVision, San Francisco, CA, USA), a cell permeable caspase 12 inhibitor. BMS345541 (1–5 μM, #16667, Cayman, Ann Arbor, MI, USA), a cell permeable IKKα/β inhibitor. Multiple antibodies were purchased and applied for either immunofluorescence staining, western blotting, or coimmunoprecipitation. Antibodies for a variety of proteins are listed as follows: antibody for Casp12 (#55238-1-AP, Proteintech, Rosemont, IL, USA), IκBα, IKKα’IKKβ, p-IκBα, and p-p65 (#4812, #2682, #8943s, #9246, #3033, Cell Signaling, Danvers, MA, USA), IKKα, NEMO (#sc14A231, #sc8032, Santa Cruz, Dallas, Texas, USA), or GFP tags (#GTX628528, GeneTex, Hsinchu City, Taiwan). P-IkB Human Casp12 (hCasp12), Casp12 (GFP-tagged), is from the human Casp12 cDNA (#RG231175, OriGene, Rockville, MD, USA).

### 4.2. Site-Directed Mutagenesis of pC12

ΔCasp12-Q (VC12) was harbored from pC12 lacking catalytic Cys residue. The QACRG pentapeptide motif containing the active site cysteine was mutated. Site-directed mutagenesis was used to introduce conservative cysteine to alanine substitutions at the positions in the full-length Casp12 cDNA. Site-directed mutagenesis of the 3′UTR-luciferase reporter vectors was carried out using PCR methods. The PCR reaction mixture contained 1 μg pCMV-Casp12, 12.5 μL 2× Extensor Hi-fidelity PCR Master Mix (ABgene, Portsmouth, NH, USA), and 0.1 μg of each primer. The oligonucleotide primers were presented: 5′-CAAGGTCATCATCATGCAAGCCGCCCGAGGCAATGGTGCTGGGATTG-3′ (sense) and 5′-CAATCCCAGCACCATTGCCTCGGGCGGCTTGCATGATGATGACCTTG-3′ (antisense). PCR was performed at the cycle of 95 °C (30 s), 55 °C (60 s), and 68 °C (8 min) for 12 cycles. The methylated parental DNA was then digested with 20 U *Dpn*I (NEB, Schwalbach, Germany) at 37 °C and then constructed into *pEGFP**-*N3** expression vector. The mutation fragment was transformed into JM109 cells, which can repair the nicked DNA. The presence of ΔCasp12-Q was verified by direct DNA sequencing.

### 4.3. Transient Transfection and Luciferase Reporter Assays

Transient transfection of plasmids was achieved using Lipofectamine 2000 (Invitrogen, Carlsbad, CA, USA). The plasmids introduced to NPC cells include pC12 plasmids and/or NF-κB reporter (Luc) vector containing five copies of a NF-κB response element (NF-κB-RE) and pSV-β-galactosidase control vector (Promega, Madison, WI, USA). The transfected cells were treated with 10 ng/mL TNF-α (PeproTech, Rocky Hill, NJ, USA) for 6 h. The cell lysates were collected 24 h post-transfection. The samples were then subjected to luciferase activity assay (Promega) to quantify the NF-κB activity levels. To confirm the transcription efficiency, the samples were tested and standardized with β-galactosidase enzyme assay (Promega).

### 4.4. Immunoprecipitation and Western Blot

The transfected NPC cells were lysed and collected 24 h post-transfection with Mammalian Protein Extraction Reagent (M-PER; Pierce Chemical Co., Dallas, TX, USA) and NE-PER Nuclear and Cytoplasmic Extraction Reagents kit (Thermo Fisher Scientific, Waltham, MA, USA). The protein in cytosolic and nuclear extracts were separated with SDS-PAGE gel electrophoresis. The result was transferred onto polyvinylidene difluoride membranes (Immobilon (TM)-P, Millipore, Burlington, MA, USA). The western blots were probed with the indicated primary antibodies and the corresponding horseradish peroxidase conjugated secondary antibodies. For immunoprecipitation, the cell lysates were prepared with PureProteome Protein A/G Mix Magnetic Beads (LSKMAGAG10, EMD Millipore, Darmstadt, Germany) then subjected to western blot. The immunoreactive bands were analyzed by a densitometer.

### 4.5. Statistical Analyses

All statistical data are presented as mean ± SD. Comparison of protein levels between two independent groups was done with the Kruskal–Wallis test. Differences between proteins were identified by Dunn’s multiple comparisons test. The pre- and post-treatment protein levels in each cell line were analyzed by the Mann–Whitney U test. *p* < 0.05 was set as the significance threshold.

## Figures and Tables

**Figure 1 ijms-22-04610-f001:**
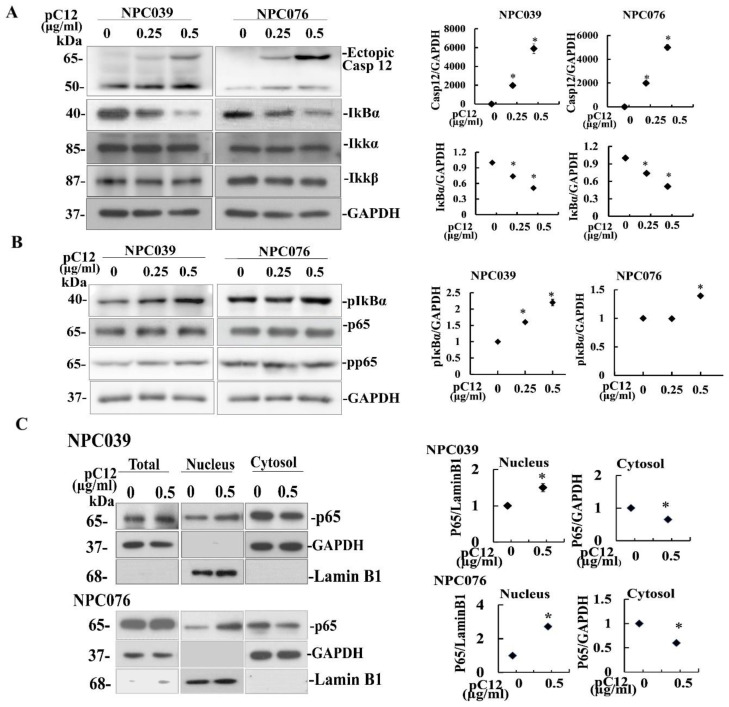
Casp12 decreased the IκBα protein and the p65 nuclear translocation. NPC cells were transfected with pC12 (0–0.5 μg/mL) for 24 h. (**A**,**B**) Ectopic hCasp12 decreased the expression level of IκBα protein, but increased the phosphorylated level of IκBα. The total cell lysates were subjected to western blot, and underwent immunoblotting for the indicated antibodies (Casp12, IκBα, IKKα, IKKβ, pIκBα, pp65, p65, and GAPDH). Endogenous Casp12 (45 kDa) and GFP-hCasp12 (~65 kDa) were detected with the anti-Casp12 antibody. The blot shown is one representative of three independent experiments and one replicate per treatment. Densitometric analysis of the detected protein immunoblot results normalized to GAPDH. Values are relative to un-transfected (Cnt) and are plotted as arbitrary units. (**C**) Ectopic expression of Casp12 induced the p65 nuclear translocation. NPC cells were transfected with hpC12 (0–0.5 μg/mL) for 24 h and then the cytoplasmic fraction and nuclear fraction underwent western blot analysis with indicated antibodies. GAPDH was shown as the cytoplasmic loading control. Lamin1 was shown as the nuclear loading control. The data from three independent experiments (*n* = 3) are graphed as the mean ± s.e.m. Data from each experimental group were compared with the control group by analysis of variance. * *p* < 0.05.

**Figure 2 ijms-22-04610-f002:**
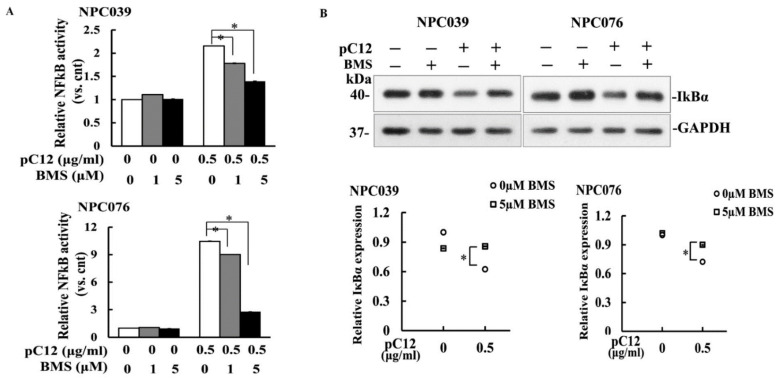
The hpC12 effects were blocked by the IKK inhibitor. NPC cells were transfected with pC12 plasmid, NF-κB reporter plasmids, and pSV-β-galactosidase vector in the presence of the IKK inhibitor (BMS, 0–5 μM) for 24 h. (**A**). The cell lysates were used to detect the luciferase activity. Luciferase activity is presented relative to the control group. (**B**). The protein level of IκBα was markedly ameliorated by BMS. The pC12-transfected cell lysates were subjected to western blot with anti-IκBα antibody. Values of densitometric analysis relative to control group are plotted as arbitrary units. The data from three independent experiments are graphed as the mean ± s.e.m. (*n* = 3), * *p* < 0.05.

**Figure 3 ijms-22-04610-f003:**
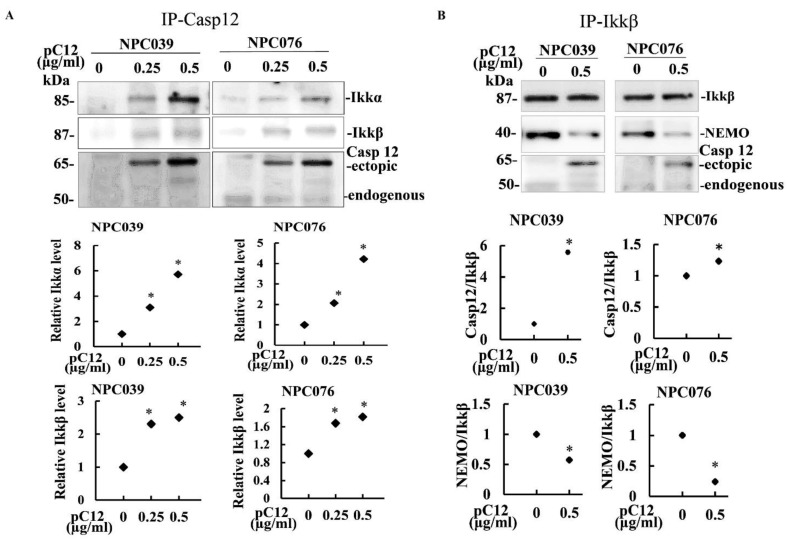
The physical interaction between hCasp12 and the IKK complex. NPC cells were transfected with pC12 for 24 h. (**A**,**B**) The physical interaction of pC12 and IKKα/β. The immunoprecipitates were validated by immunoprecipitation with anti-hCasp12 antibody (**A**) and anti-IKKβ antibody. The immunoprecipitates were subjected to western blot to detect the protein levels of IKKα/β and hCasp12. Values of densitometric analysis relative to control group are plotted as arbitrary units. The data from three independent experiments are graphed as the mean ± s.e.m. (*n* = 3), * *p* < 0.05.

**Figure 4 ijms-22-04610-f004:**
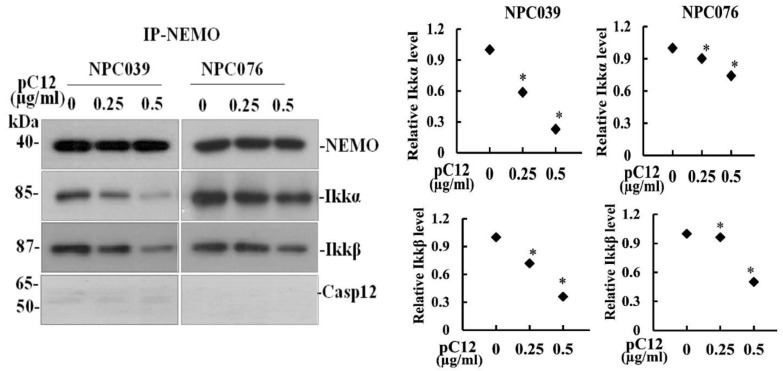
hCasp12 did not physically interact with NEMO. NPC cells were transfected with pC12 for 24 h. The transfected cell lysates were immunoprecipitated with anti-NEMO antibody. The immunoprecipitates underwent western blot analysis with indicated antibodies. The data from three independent experiments are graphed as the mean ± s.e.m., * *p* < 0.05.

**Figure 5 ijms-22-04610-f005:**
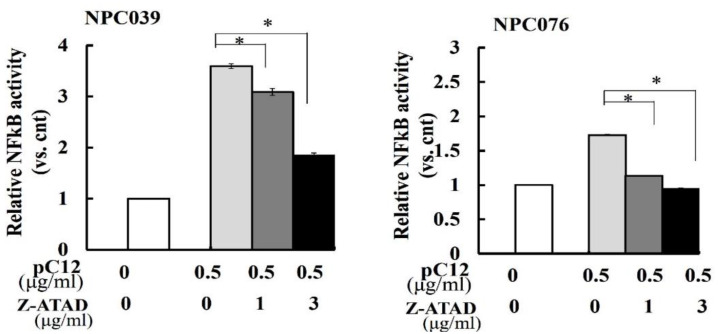
The hCasp12-mediated NF-κB activity inhibited by Casp12 inhibitor. NPC cells were transfected with pC12 plasmid, NF-κB reporter/pSV-β-galactosidase plasmids in the presence of hCasp12 inhibitor (z-ATAD-fmk) for 24 h. The cell lysates were used to detect the luciferase activity. Results are presented as the ratio of luciferase to β-gal activities normalized to untreated cells. The data from three independent experiments are graphed as the mean ± s.e.m., * *p* < 0.05.

**Figure 6 ijms-22-04610-f006:**
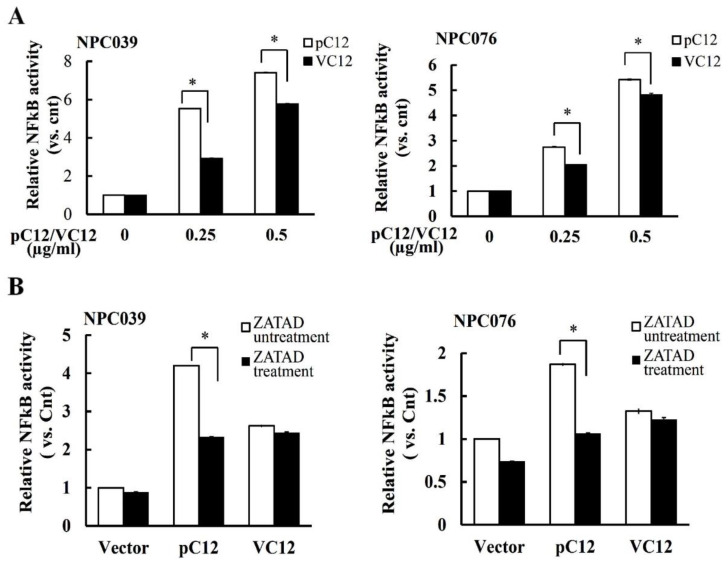
The active site of the hCasp12 in the regulation of NF-κB. (**A**) NPC cells were transfected with pC12/VC12 (0~0.5 μg/mL), (**B**) NPC cells were transfected with pC12/VC12 (0.5 μg/mL), NF-κB reporter/pSV-β-galactosidase plasmids in the presence of ATAD-fmk (3 μg/mL) for 24 h. The transfected cell lysates were used to detect the luciferase activity. The data from three independent experiments are graphed as the mean ± s.e.m. (*n* = 3), * *p* < 0.05.

**Figure 7 ijms-22-04610-f007:**
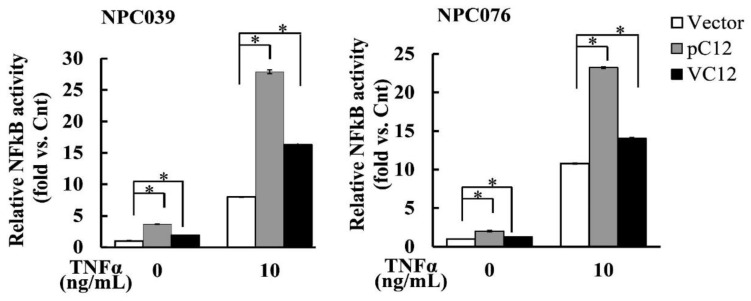
The Casp12-mediated NF-κB activity enhanced by TNFα stimuli. NPC cells were transfected with pC12/VC12 plasmid, NF-κB reporter/pSV-β-galactosidase plasmids for 18 h. The transfected cells were treated with TNF-α (10 ng/mL) for 6 h and the cell lysates were used to detect the luciferase activity. Transfection with an empty vector (pEGFP-N3) plasmid (0.5 μg/mL) was as a control group cells. Luciferase activity is presented relative to the control group. The data from three independent experiments are graphed as the mean ± s.e.m. (*n* = 3). * *p* < 0.05.

## Data Availability

All data generated or analyzed in this study are available from the corresponding author on reasonable request.

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
