# Peer review of "Human Caspase 12 Enhances NF-κB Activity through Activation of IKK in Nasopharyngeal Carcinoma Cells"

_ijms, 2021, doi:10.3390/ijms22094610_

Round 1

Reviewer 1 Report

This manuscript investigated Casp12 cDNA (pC12) and the effect of human Casp12 (hCaso12) on the NF-κB activity in NPC cells. The authors indicated that hCasp12 enhanced NF-κB activity through activation of IKK in NPC cells. This is an interesting observation that may indicate hCasp12-mediated inflammation might reveal new therapeutic targets for nasopharyngal carcinoma. However, I have the following suggestions to the authors with further revision for the manuscript.

  1. The authors should modify the order of the references number. For example, No.17, and I cannot find the No.19 reference.
  2. In Figure 2 legend: b.→
  3. In figure 3 legend: a/b.→A/B, c.→ And In figure 3, I cannot find figure 3C.

Author Response

Reviewer 1

Comments and Suggestions for Authors

This manuscript investigated Casp12 cDNA (pC12) and the effect of human Casp12 (hCaso12) on the NF-κB activity in NPC cells. The authors indicated that hCasp12 enhanced NF-κB activity through activation of IKK in NPC cells. This is an interesting observation that may indicate hCasp12-mediated inflammation might reveal new therapeutic targets for nasopharyngal carcinoma. However, I have the following suggestions to the authors with further revision for the manuscript.

  1. The authors should modify the order of the references number. For example, No.17, and I cannot find the No.19 reference.
  2. In Figure 2 legend: b.→
  3. In figure 3 legend: a/b.→A/B, c.→ And In figure 3, I cannot find figure 3C.

Response:

Thanks for the comments

  1. The reference order has been modified as Reviewer’s suggestion.

2~3. It has been changed into “capital A/B/C”and shown in Fig. 2~Fig. 3 legend.

The legend of Fig. 3C has been deleted.   

Reviewer 2 Report

The authors investigated the effect of hCasp12 on NF-KappaB activation in two NPC cell lines. They found that hCasp12 transfection led to a decrease in iKappaB levels and p65 nuclear translocation as well as increased NF-KappaB reporter activity. This was dependent of the Casp12 catalytic activity as a mutant devoid of the casp12 catalytic domain was less efficient in activating NF-KB pathway.

In principle the study is well conceived however this reviewer cannot recommend it for publication because of the lack of:

  • functional assays on how the activation of NF-KB pathway by casp12 would affect proliferation and invasion
  • more physiological cellular models that would not be based on the ectopic expression of casp12 on only two NPC cell lines

Author Response

#Reviewer 2

Comments and Suggestions for Authors

The authors investigated the effect of hCasp12 on NF-KappaB activation in two NPC cell lines. They found that hCasp12 transfection led to a decrease in iKappaB levels and p65 nuclear translocation as well as increased NF-KappaB reporter activity. This was dependent of the Casp12 catalytic activity as a mutant devoid of the casp12 catalytic domain was less efficient in activating NF-KB pathway.

In principle the study is well conceived however this reviewer cannot recommend it for publication because of the lack of:

  • functional assays on how the activation of NF-KB pathway by casp12 would affect proliferation and invasion

 Response:

 Induction of Casp12 is implicated in NPC cell invasion after the proinflammation stimuli [1]. The functional assay on how the activation of NF-κB pathway by casp12 would affect NPC cell invasion has been validated in this study. The following statement has been shown in Introduction section (line 54-line 60, page 2). “Induction of Casp12 by PMA (a PKC analogue) leads to cell invasion in NPC cells [1]. Ectopic murine Casp12 protein is shown to decrease IκBα protein and significantly increase the NF-κB activity in NPC cells [1]. Moreover, depletion of Casp12 by its specific siRNA impairs the cancer cell invasion via inactivation of NF-κB [1]. Recent study indicates knockout of Casp12 fails to induce the Casp1 activation and IL-1β and IL-18 production [2]. These observations suggested a role of Casp12 in NF-κB-mediated cell invasion. “

  • more physiological cellular models that would not be based on the ectopic expression of casp12 on only two NPC cell lines

Response:

   The previous study has shown Casp12 is implicated in the regulation of NF-κB-mediated cellular invasion via the modulation of IκBα protein in NPC cells [1]. Induction of Casp12 is implicated in NPC cell invasion after the proinflammation stimuli [1]. Further study was to explore the molecular mechanism of hCasp12 in inflammation-mediated cell invasion. To investigate the effect mechanism of hCasp12 in the regulation of NF-κB activity, NPC cells were transfected with the full length of human Casp12 cDNA (pC12). One of the reliable quantitative methods measuring NF-κB activity is by transfection with ectopically expressed hCasp12 gene. In our opinion, it is reasonable to explore the molecular mechanism of hCasp12 in the activation of NF-κB by utilization of two NPC cell lines.   

  1. Chu, W.-K.; Hsu, C.-C.; Huang, S.-F.; Hsu, C.-C.; Chow, S.-E. Caspase 12 degrades IκBα protein and enhances MMP-9 expression in human nasopharyngeal carcinoma cell invasion. Oncotarget 2017, 8, 33515-33526, doi:10.18632/oncotarget.16535.
  2. Vande Walle, L.; Jiménez Fernández, D.; Demon, D.; Van Laethem, N.; Van Hauwermeiren, F.; Van Gorp, H.; Van Opdenbosch, N.; Kayagaki, N.; Lamkanfi, M. Does caspase-12 suppress inflammasome activation? Nature 2016, 534, E1-E4, doi:10.1038/nature17649.

Round 2

Reviewer 2 Report

The authors unfortunately did not address my points at all.

This reviewer would like to see an experiment where hCasp12 in the experimental settings used by the authors would induce cell migration although already shown in the literature these experiment would validate the system used in this study and reinforce and otherwise weak finding

In addition I would like to ask the authors whether, beside the ectopic expression, there would be another way  (CRISPR gene activation for instance) to induce hcasp12 expression. Another concern is how the level of overxpressed hcasp12 would reflect the physiological levels.

Without experimentally addressing these points I cannot recommend this study for publication.